# Reduced Graphene Oxide Aerogel inside Melamine Sponge as an Electrocatalyst for the Oxygen Reduction Reaction

**DOI:** 10.3390/ma14020322

**Published:** 2021-01-09

**Authors:** Roman A. Manzhos, Sergey A. Baskakov, Evgeny N. Kabachkov, Vitaly I. Korepanov, Nadezhda N. Dremova, Yulia V. Baskakova, Alexander G. Krivenko, Yury M. Shulga, Gennady L. Gutsev

**Affiliations:** 1Institute of Problems of Chemical Physics, Russian Academy of Sciences, 142432 Chernogolovka, Russia; rmanzhos@yandex.ru (R.A.M.); baskakov@icp.ac.ru (S.A.B.); en.kabachkov@gmail.com (E.N.K.); dremova@icp.ac.ru (N.N.D.); ybaskakova@yandex.ru (Y.V.B.); krivenko@icp.ac.ru (A.G.K.); yshulga@gmail.com (Y.M.S.); 2Chernogolovka Scientific Center, Russian Academy of Sciences, 142432 Chernogolovka, Russia; 3Institute of Microelectronics Technology and High Purity Materials, Russian Academy of Sciences, 142432 Chernogolovka, Russia; korepanov@iptm.ru; 4Institute of New Materials and Nanotechnologies, National University of Science and Technology MISIS, Leninsky pr. 4, 119049 Moscow, Russia; 5Department of Physics, Florida A&M University, Tallahassee, FL 32307, USA

**Keywords:** reduced graphene oxide aerogel, melamine sponge, oxygen reduction reaction, lyophilization, composite material, linear sweep voltammetry

## Abstract

A graphene oxide aerogel (GOA) was formed inside a melamine sponge (MS) framework. After reduction with hydrazine at 60 °C, the electrical conductive nitrogen-enriched rGOA-MS composite material with a specific density of 20.1 mg/cm^3^ was used to fabricate an electrode, which proved to be a promising electrocatalyst for the oxygen reduction reaction. The rGOA-MS composite material was characterized by elemental analysis, scanning electron microscopy, X-ray photoelectron spectroscopy, and Raman spectroscopy. It was found that nitrogen in the material is presented by different types with the maximum concentration of pyrrole-like nitrogen. By using Raman scattering it was established that the rGOA component of the material is graphene-like carbon with an average size of the sp^2^-domains of 5.7 nm. This explains a quite high conductivity of the composite obtained.

## 1. Introduction

Aerogels of graphene oxide (GO), reduced graphene oxide (rGO), and composites on their base are being actively studied [1,2,3,4,5,6,7,8,9,10,11,12,13,14,15,16,17]. These materials are of considerable interest due to their low specific density (0.16 mg/cm^3^ [1]) and high specific surface area (up to 960 m^2^/g [2]). Aerogels with high conductivity (up to 500 S/m [5]) can be obtained by using special treatments and/or adding additives [3,4,5]. Modified aerogels can be used as catalysts, adsorbents of heavy metals, oils, and dyes, as well as sensors, electrodes for batteries and supercapacitors [7,8,9,10,11,18,19,20,21,22,23]. In addition, rGO aerogels are also used as materials for electrocatalysis [24,25,26].

From the point of view of practical applications, graphene-like materials have attracted great interest as catalysts for the oxygen reduction reaction (ORR) in fuel cell cathodes [27,28,29,30]. Reduced graphene oxide can be used as a carrier for catalyst nanoparticles composed of Pt and Pd as well as of their alloys [31,32,33] or transition metal oxides [28,34,35,36]. In addition, the graphene structures demonstrated excellent electrocatalytic activity towards ORR when doped with *p*-elements (N, S, P, and B) [27,28,30]. One of the most popular methods for obtaining N-doped graphene structures is the reduction of graphene oxide with hydrazine or ammonia [29,34,35,37]. In our previous work [38], we performed the reduction of a graphene oxide aerogel in hydrazine vapors and obtained conductive aerogel with a nitrogen content of 3 wt% and significant electrocatalytic activity towards ORR.

However, our experience has shown that the applicability of rGO aerogels without crosslinking (binding) agents is often limited by their fragility. Our experience is supported by the literature data. For example, graphene foam synthesized by a template chemical vapor deposition method is brittle and prone to fracture upon the removal of a thicker metal template [39]. Aerogels obtained by the hydrothermal or freeze-drying methods do crack due to the different heating or cooling rates of the outer and inner parts [23,40].

Crosslinking agents should improve the mechanical properties of aerogels, but they can alter the catalytic or absorption characteristics of rGO. It is interesting to produce open-frame aerogels with suitable mechanical characteristics as an alternative solution. Suitable frameworks must have a low density and be chemically inert under ambient conditions. In principle, these criteria are met by melamine sponge, which has a high porosity of over 99%, pore sizes of approximately 100 µm [41], low density, excellent mechanical properties, and, most importantly, it is a cheap and ecologically clean product [41,42,43,44]. The trademark “melamine sponge” usually means a melamine-formaldehyde resin sponge whose chemical formula is presented in the Appendix A. Currently, there are a large number of publications in the literature describing new perspective applications of melamine sponge after its modification.

There are several works in which a melamine sponge was used in combination with graphene oxide or reduced graphene oxide [45,46,47,48,49,50]. In particular, the melamine/graphene sponge (M/GS) was prepared by a dip-coating method and low-temperature thermal reduction [45]. This M/GS was immersed into the carbon black/tetrahydrofuran (THF) dispersion and dried in air. In [46], an oil/water separation material with robust mechanical properties was developed by modifying a melamine sponge with silk fibroin-graphene oxide. The surface of the melamine sponge (MS) was also modified with polydopamine and then coated with glutathione/graphene oxide as an adsorbent for the removal of Pb(II) from aqueous solutions [47]. The superhydrophobic-reduced graphene oxide-modified melamine sponge may also be prepared via a facile ultrasonic-microwave synergistic method [48]. This material has shown high selectivity for trapping various oils and organic solvents from water, as well as excellent recyclability and stability. A significant part of the work on obtaining hydrophobic composites based on reduced graphene oxide and melamine sponge was aimed at separating water/oil mixtures (see, for example, [49,50]). To the best of our knowledge, there is no work in the literature where a graphene oxide aerogel was grown in the pores of a melamine sponge, and the GOA-MS composite reduced in hydrazine vapors was used as an electrocatalyst for the oxygen reduction reaction.

The present paper describes a method for preparing GO aerogel in open pores of a melamine sponge, thus forming a GOA-MS composite. Subsequent reduction with hydrazine vapors results in a rGOA-MS composite, which was characterized by elemental analysis, scanning electron microscopy (SEM), X-ray photoelectron spectroscopy (XPS), and Raman spectroscopy. One can anticipate that this composite will have many perspective applications. In this work, we describe the use of the rGOA-MS composite as a catalyst for the electroreduction of oxygen.

## 2. Experimental

Melamine sponge with a density of 9.8 mg/cm^3^ was manufactured by the JSC Accent (Saint Petersburg, Russia) from raw material from BASF SE (Ludwigshafen, Germany). For the GO synthesis, we used the modified Hummers method [51]. The GO production is described in detail in [52,53] and Appendix A. The sponge was used in our experiments without additional pre-treatments. First, the sponge was impregnated by immersion into GO suspension (17 mg/mL) in distilled water. Next, the sponge was removed from the suspension and was slowly frozen on a copper plate cooled with liquid nitrogen. The frozen sponge was placed in a glass vessel with a silicone cup and lyophilized in a freeze dryer (model FDS5512 from IlShin BioBase, Dongducheon, Korea). The dried aerogel GOA-MS had a density of 26.7 mg/cm^3^. The aerogel was treated with hydrazine vapor to reduce GO, namely a portion of GOA-MS was placed in a sealed plastic vessel together with a portion of hydrazine hydrate (4–5 mL in a Petri dish). The vessel was thermostated at 60 °C for 48 h. The resulting reduced aerogel had a black color (Figure 1B), and its density was 20.1 mg/cm^3^.

The elemental analysis of the samples for the content of C, H, N, and S was carried out on an analyzer, the “VarioMicrocube” Elementar GmbH (Hanau, Germany). Raman spectra were recorded with the use of the Bruker Senterra micro-Raman system. The excitation wavelength was 532 nm and the laser power was ~1 mW at the sample point with a beam waist of ~1 µm. The contact water-wetting angle was measured on an OCA 20 instrument (Data Physics Instruments GmbH, Filderstadt, Germany) at room temperature. Electron micrographs were obtained with a JEOL JSM-5910LV scanning electron microscope (JEOL Ltd., Tokyo, Japan) (electron energy 20 kV, chamber pressure 2 × 10^−5^ Pa).

XPS spectra were obtained using a Specs PHOIBOS 150 MCD electron spectrometer (SPECS GmbH, Berlin, Germany) with a Mg cathode (*hν* = 1253.6 eV). The vacuum in the spectrometer chamber did not exceed 4 × 10^−8^ Pa. The survey spectra were recorded in the constant transmission energy mode of 40 eV whereas the individual lines were recorded with 10 eV. The survey spectra and individual lines were recorded in 1.00 eV and 0.03 eV increments, respectively. The background subtraction was carried out according to the Shirley method [54], and the spectra decomposition into mixed Gaussian–Lorentz peaks was performed using the Casa XPS software (version 2.3.19, Casa Software Ltd, Teignmouth, UK). Quantification of atomic content was done using the sensitivity factors from the CasaXPS elemental library.

The linear sweep voltammetry (LSV) was performed in a three-electrode cell using the set-up with a RRDE-3A rotating disk electrode (ALS Co., Ltd., Tokyo, Japan) and an Autolab PGSTAT302N (Metrohm Autolab, Utrecht, The Netherlands) in an O_2_-saturated solution of 0.1 M KOH at a potential scan rate of *v* = 10 mV/s and electrode rotation rates ω = 360–6400 rpm. The analysis of voltammetry results was carried out within the Koutecký–Levich equation [55]:(1)$$\frac{1}{j}=\frac{1}{j_{k}}+\frac{1}{j_{d}}$$
where *j*_k_ = *n*F*kc*^0^ and *j*_d_ = 0.62*n*FD^2/3^ω^1/2^*υ*^−1/6^*c*^0^ are the kinetic current density and the limiting diffusion current density, respectively, F is the Faraday constant (F = 96,485 C/mol), *n* is the number of electrons participating in the electrode reaction, *k* is the oxygen reduction rate constant, D is the coefficient of oxygen diffusion in 0.1 M KOH (D = 1.9 × 10^−5^ cm^2^/s), *υ* is the kinematic viscosity of 0.1 M KOH (*υ* = 0.01 cm^2^/s), and *c*^0^ is the bulk concentration of dissolved oxygen (*c*^0^ = 1.2 × 10^−3^ M in 0.1 M KOH) [56,57].

The rGOA-MS composite was ground in liquid nitrogen in a nephrite jade mortar and the resulting powder was dispersed in water using a powerful ultrasonic disperser. A glassy carbon (GC) disk of 3 mm in diameter pressed into PEEK polymer was used as a working electrode. The surface of the initial GC electrode was polished with 1 µm of Al_2_O_3_ powder. Then, ca. 5 µL of the rGOA-MS suspension (0.5 mg/mL) with ca. 0.01 wt% Nafion polymer added was drop-casted on the GC surface and dried at ambient temperature. Platinum wire with a 1 cm^2^ surface area was an auxiliary electrode; a reference electrode was represented by an Ag/AgCl (saturated KCl) electrode. All potentials (*E*) were converted to the reversible hydrogen electrode (RHE) scale (*E*_RHE_ = *E*_(Ag/AgCl)_ + 0.964 V).

## 3. Results and Discussion

### 3.1. Elemental Analysis

The data on our sample compositions are presented in Table 1. Minor amounts of sulfur can be present in GO obtained by the Hummers technique (see, for example, [58]). The technological impurities of melamine sponge may also contain sulfur. The GO content in the composite can be estimated by taking into account the fact that GO contains negligible amount of nitrogen. This gives 74 wt% of GO in GOA-MS. The tests show that the hydrazine vapor treatment induces no change in the melamine sponge. Therefore, chemical changes upon the transition from GOA-MS to rGOA-MS should be attributed to GO only. Table 1 shows a pronounced reduction of GO at this stage (the oxygen content decreases). Nitrogen content in rGOA-MS becomes significantly higher. High nitrogen content may indicate good performance of the material as an electrocatalyst for oxygen reduction [59]).

### 3.2. SEM

The SEM images of the initial MS and rGOA-MS composite are shown in Figure 1. It may be seen that the average pore size in MS is equal to approximately 100–200 µm, and a relatively loose rGO substance is uniformly distributed inside the melamine sponge framework.

### 3.3. Raman Spectra

Micro-Raman spectroscopy makes it possible to study the areas with predominant distributions of rGO and MS separately. The spectra for two different spots in the rGOA-MS composite are shown in Figure 2. In spectrum *1* corresponding to the MS, the most intense peak at 985 cm^−1^ belongs to the CNC+NCN bending vibration of the triazine ring [60,61,62,63,64]. In spectrum *2,* two dominant features are the D and G bands of the graphene-like carbon [65] and wide band at ~2700 cm^−1^. For graphene, the 2D peak is typically observed at 2700 cm^−1^. For multi-layer graphene, the position may be shifted to lower energies. For GO, the typical position is close to that of a graphene monolayer [66], but the width is significantly higher (FWHM ~200 cm^−1^ as compared to 24 cm^−1^ for the graphene monolayer).

The Raman spectrum of rGO was deconvoluted into five peaks as it was previously suggested [67]. Prior to deconvolution, the background was subtracted by the algorithm described elsewhere [68]. The peak shapes were approximated with the flexible pseudo-Voigt profiles [69]. The positions, half-widths, and relative intensities of the resulting fitting functions are shown in Table 2.

The ratio between the D and G band intensities (*I_D_/I_G_*) can serve as a disorder measure in the carbon lattice. The *I_D_/I_G_* values can also be used to evaluate the size of the *sp*^2^-domains *L_a_* [65] as follows:
*L_a_* = (2.4 × 10^−^^10^) λ^4^_L_ (*I_D_/I_G_*)^−^^1^(2)
where λ_L_ is the wavelength (in nm) of the excitation laser. The domain size *L_a_* estimated from Equation (2) equals to 5.7 nm for the rGOA-MS material.

### 3.4. XPS

Figure 3 shows the survey XPS spectra of the rGOA-MS sample. The elemental composition of the surface, calculated using the integral intensities of the analytical lines (marked in the figure), corresponds to formula C_0.88_O_0.08_N_0.03_S_0.01_. The surface composition is different from that of the bulk (see Table 1). In particular, the surface has lower nitrogen and oxygen concentrations. The nitrogen depletion at the surface may indicate a low electrocatalytic efficiency of rGOA-MS for oxygen reduction. However, this result is not conclusive because the surfaces accessed in the XPS experiment and involved into the redox reaction may be different.

The C 1*s* line shape and its deconvolution into six symmetric pseudo-Voigt profiles is displayed in Figure 4A. The binding energies and peak intensities are presented in Appendix A (see Appendix A). The dominant peak is located at 284.6 eV, which is typical for *sp*^2^ carbon materials. The second intense peak at 285.9 eV can be assigned to the carbon atoms single-bonded to oxygen or nitrogen, i.e., to hydroxy- and epoxy-groups, respectively, and also to pyridine and pyrrole nitrogen. The contributions of these groups cannot be reliably separated.

The N 1*s* structure allows one to identify the presence of surface nitrogen-containing groups. Nitrogen in graphite-like matrix can exist [70] in the following configurations: Pyridine-like (N1), pyrrole (5-membered cycle) (N2), graphite-like N (N3, N4), and oxidized pyridine N (N5). The N1 and N2 atoms are located at the edges of the graphene layers or at the defect sites. The graphite-like N3 and N4 substitute carbon atoms in the graphene layer at the edges (N3) and in the center (N4). The pyridine-like N1 can present also in the oxidized form (N5).

The XPS peak positions for these nitrogen types are well known (see [70] and references therein). N1 corresponds to a peak at 398.0–399.3 eV, N2 corresponds to a peak around 399.8–401.2 eV, N4 corresponds to a peak close to 401 eV, and N3 corresponds to a peak around 402.3 eV. The highest energy of 402.8 eV has a peak corresponding to the oxidized pyridine type N4.

In the sample studied, the most intense N 1*s* peak is located at 400.1 eV (see Figure 4B and Appendix A). This main peak can belong to pyrrole-like (N2) as well as to the graphite-like (N4) structures. A minor peak (with about 10% of the total N 1*s* intensity), which can be distinguished at 399.1 eV, can safely be assigned to the pyridine-like nitrogen (N1). In the previous work [71], based on the use of the model catalysts, it was shown that pyridine-like nitrogen plays the role of an active center for oxygen reduction under acidic conditions.

### 3.5. Oxygen Reduction Reaction

The linear sweep voltammetry carried out in a solution of 0.1 M KOH saturated with oxygen for the glassy carbon electrode loaded with rGOA-MS as well as for the initial GC are presented in Figure 5 (the electrode rotation rate was 2000 rpm and the potential scan rate was 10 mV/s). Based on the results of processing voltammograms measured at different electrode rotation rates (Figure 6A), from the slope of Koutecký–Levich (KL) plots (Figure 6B) and in accordance with Equation (1), the number of electrons *n* participating in the oxygen reduction reaction were defined at various potentials in Figure 7. The inset in Figure 7 depicts Tafel plots for the investigated objects. The slopes of lines are 50.3 mV/dec and 33.8 mV/dec for the initial GC and rGOA-MS, respectively, which are consistent with the values reported in the literature [72,73]. As can be seen from Figure 5, the GC electrode coated with rGOA-MS demonstrates somewhat higher oxygen reduction currents than the initial GC. On the LSV curve for rGOA-MS (curve *2* in Figure 5), two distinct waves can be distinguished, which correspond to the reduction of oxygen to hydrogen peroxide in the potential range from 300 mV to 650 mV (n ≈ 2) and water at *E* < 100 mV (*n* ≈ 3.4–3.5). Thus, a noticeable catalytic effect towards the oxygen reduction reaction is observed for rGOA-MS, namely, there is a ca. 60 mV decrease in the ORR overpotential and higher values of *n* are observed for rGOA-MS at *E* < 300 mV as compared to glassy carbon. The results obtained are less prominent as it was for a reduced graphene oxide aerogel [38], which demonstrated a ca. 90 mV decrease in ORR overpotential and higher values of the electron transfer number (n ≈ 2.4–2.8 in the potential range from 350 mV to 550 mV). Evidently, the observed effect is rather small, which is caused by a low concentration of active centers (surface defects, edge regions of graphene-like structures, and dopant atoms embedded into graphene layers) on the surface of rGOA-MS for the adsorption of both O_2_ and intermediates of its reduction. It can be assumed that in accordance with [23], insufficient catalytic activity of rGOA-MS is due to the low surface concentration of pyridine-like nitrogen. The thermal treatment of the composite could likely increase the surface concentration of nitrogen. Moreover, the melamine sponge in the composite also affects the total concentration of active centers, which is lower than that for a reduced graphene oxide aerogel [38].

## 4. Conclusions

The freeze-drying of GO in a melamine sponge framework resulted in the formation of a GOA-MS composite where the GO aerogel was uniformly distributed over the MS structure. This material was reduced with hydrazine to yield the rGOA-MS. The method of obtaining our composites differs significantly from the classical method of producing nanocarbon 3D structures by the infiltration of a porous substrate [74]. The structure of our carbon 3D material was determined not by the porous structure of the substrate, but by the ratio of water and graphene oxide of the hydrogel, as well as by the freeze-drying procedure and subsequent treatments. Elemental analysis showed that the resulting composite had a high nitrogen content of 27%, whereas the surface concentration of nitrogen estimated from the XPS spectra was significantly lower. The XPS-estimated composition of rGOA-MS corresponded to the formula C_0.88_O_0.08_N_0.03_S_0.01_. The tests showed that the rGOA-MS composite had a pronounced electrocatalytic activity towards oxygen reduction reaction. The catalytic efficiency could be enhanced by increasing the near-surface content of nitrogen.

## Figures and Tables

**Figure 1 materials-14-00322-f001:**
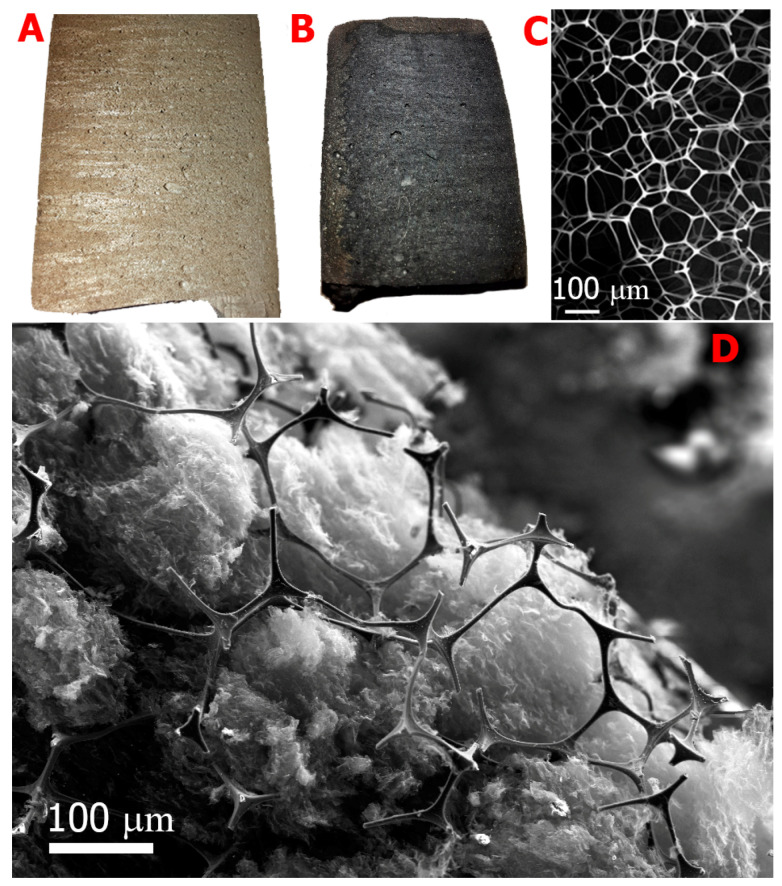
Optical images of GOA-MS (graphene oxide aerogel- melamine sponge): (**A**) rGOA (reduced graphene oxide)-MS, (**B**) SEM images of the MS, (**C**) rGOA-MS, and (**D**) samples fracture surface. The sample was fractured at the temperature of liquid nitrogen.

**Figure 2 materials-14-00322-f002:**
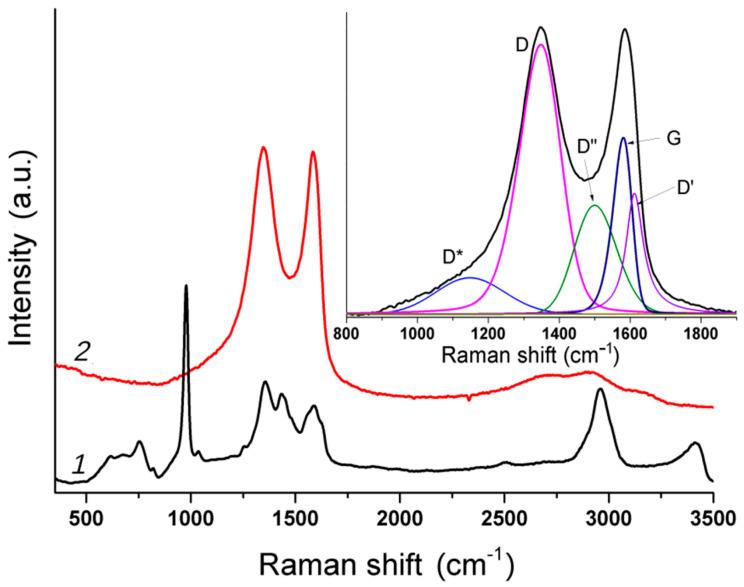
Micro-Raman spectra of the rGOA-MS composite taken from two spots with the dominant contribution of MS (*1*) and rGO (*2*). The inset shows a deconvolution of the rGO spectrum into separate peaks as suggested in [67].

**Figure 3 materials-14-00322-f003:**
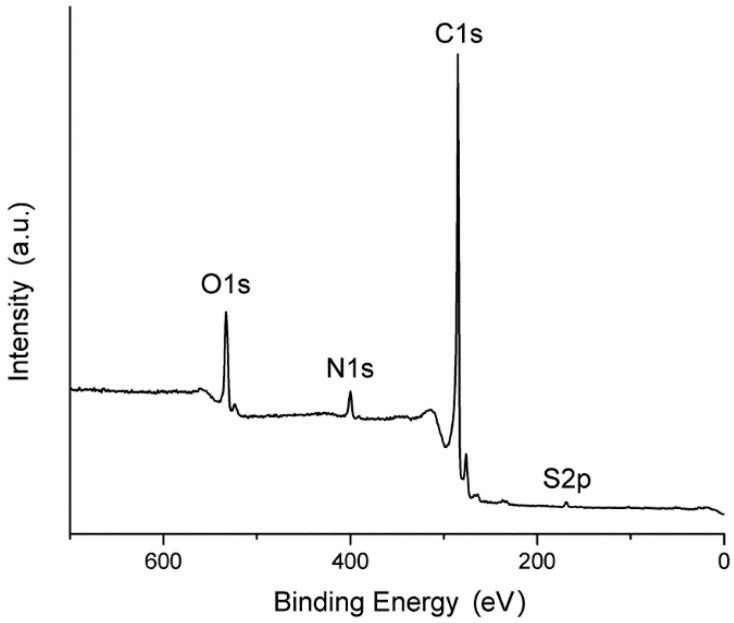
The XPS survey spectrum of the rGOA-MS sample.

**Figure 4 materials-14-00322-f004:**
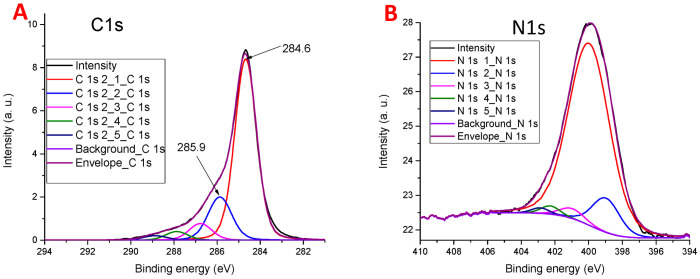
The C 1*s* (**A**) and N 1*s* (**B**) spectra of the rGOA-MS sample.

**Figure 5 materials-14-00322-f005:**
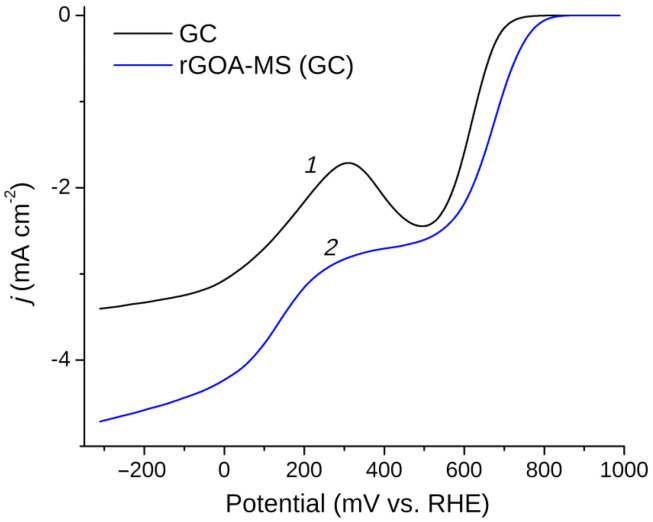
Linear sweep voltammetry in an O_2_-saturated solution of 0.1 M KOH for the glassy carbon (GC) electrode (*1*) and GC coated with rGOA-MS (*2*), *v* = 10 mV/s, ω = 2000 rpm.

**Figure 6 materials-14-00322-f006:**
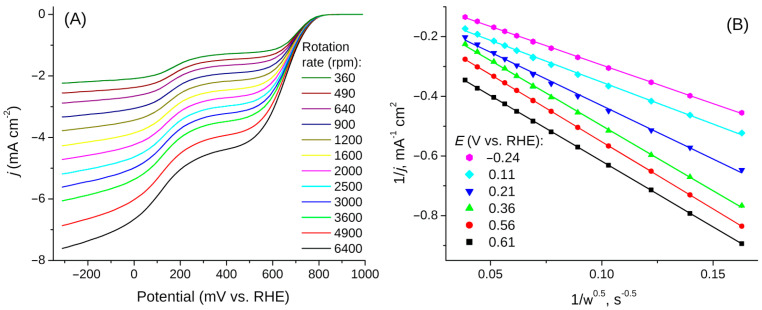
The LSVs at the GC electrode coated with rGOA-MS at different electrode rotation rates, *v* = 10 mV/s (**A**); K-L plots at selected potentials (**B**).

**Figure 7 materials-14-00322-f007:**
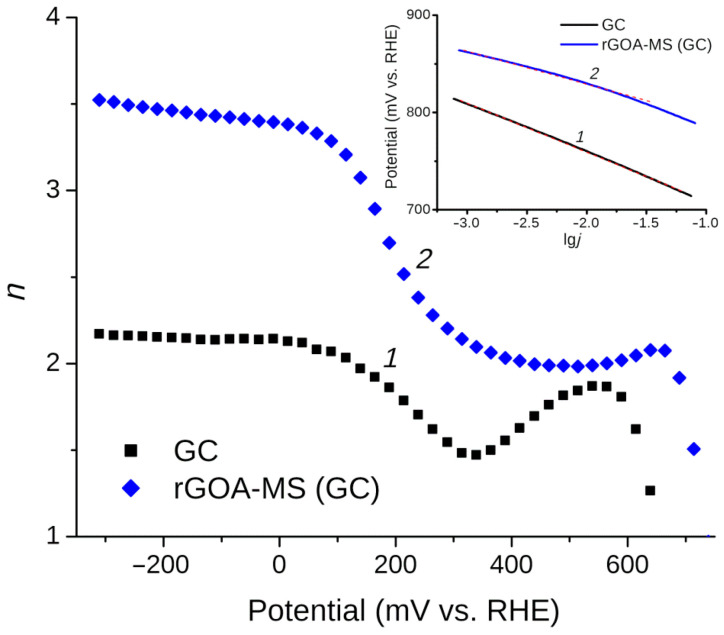
*n* vs. potential for the GC electrode (*1*) and GC coated with rGOA-MS (*2*).

**Table 1 materials-14-00322-t001:** Elemental composition (in wt%) of the samples under study. MS: Melamine sponge.

Sample	C	H	N	S	O *
MS	35.40 ± 0.22	5.164 ± 0.039	43.84 ± 0.28	1.325 ± 0.021	14.27
GOA-MS	41.65 ± 0.18	3.750 ± 0.074	17.40 ± 0.36	1.727 ± 0.058	35.47
rGOA-MS	53.34 ± 0.76	3.179 ± 0.061	27.50 ± 0.92	1.500 ± 0.106	14.48

* oxygen content was estimated by the formula [O] = 100 − $\sum_{i}[C_{i}]$, where [*C_i_*] is the content of the *i*-th element.

**Table 2 materials-14-00322-t002:** The positions (*Pos*), full widths at half maximum (*FWHM*), and intensities (*Int*) of the peaks in the Raman spectrum of rGOA-MS.

Peak	*Pos*, cm^−1^	*FWHM*, cm^−1^	*Int*, %
D*	1148	227	10
D	1348	138	46
D″	1500	140	18
G	1581	62	14
D′	1612	58	12

## Data Availability

Data available in a publicly accessible repository.

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
