# Peer review of "Reduced Graphene Oxide Aerogel inside Melamine Sponge as an Electrocatalyst for the Oxygen Reduction Reaction"

_materials, 2021, doi:10.3390/ma14020322_

Round 1
Reviewer 1 Report
In this contribution by Manzhos and co-workers, the authors characterized reduced graphene oxide aerogel inside of melamine sponge and evaluated the application potential of this solution for the oxygen reduction reaction. The study, in principle, is interesting, but the methodology contains some errors which should be eliminated to make the results credible. For now, major revision is recommended. Upon incorporation of the following suggestions given below, the article may be reconsidered for publication:
1) The introduction section is very brief (less than half a page). The authors should extend it paying particular attention to include:
- the properties of rGO for oxygen reduction reaction, which is explored in this work
- other methods to produce nanocarbon 3D structures by infiltration of porous substrates
2) Based on these two, the authors should also clearly specify the novelty factor of this contribution. What exactly is reported in this work that was not done by others?
3) There is not enough information regarding GO used in this work, which is the heart of this study. This, unfortunately, makes the study not reproducible. The authors should describe the synthesis conditions and characterize the product using at least SEM, TEM, XPS, and Raman spectroscopy. Without such information, the study is not reproducible and one cannot draw conclusions with appropriate scientific rigor.
4) Caption to Fig. 1 overflows to the following page.
5) What was the reference element used for XPS calibration?
6) Raman spectra should also be provided for the sample before the reduction.
7) Please consider using the deconvoluted curves from XPS plots to establish the relative amount of different binding configurations with respect to each other
8) The conclusions section is very short. It should also contain a description of the impact of this study and some future outlook.
Reviewer 2 Report
In this manuscript, Manzhos et al reported a facile method to produce graphene oxide aerogel inside a melamine sponge framework. Several characterization methods were used to confirm the structural/surface composition. The results indicated a materials with a high nitrogen content of 27% with pronounced electrocatalytic activity towards oxygen reduction reaction. Basically, the whole manuscript is organized and most of the experimental results are reasonable. I would recommend publishing after minor revision.
The followings are necessary to be addressed before the publication process is completed.
1) Please measure and report the specific surface area of the rGO/MS aerogels.
2) I recommend the author to add a scheme (figure) of the synthetic procedure for the preparation of the rGO aerogel. This will benefit from reader point of view.
3) As mentioned in the introduction part, GO was used to prepare many different composites. The author should discuss more about the advantages of using melamine sponge as framework. In addition, the authors can compare the performance of rGO/MS composite with other reported systems in the main text.
4) What is the pore size of the melamine sponge? Pls comment.
5) Please include chemical structures of melamine sponge (MS) and SEM micrographs of pure MS if available.
6) Page 5, Figure 2. The typical 2D peak of graphene (~2700 cm-1). Please briefly discuss the reason.
7) Please comment on the mechanical properties of the rGO/MS composite. A DMA analysis or stress/strain curve would be beneficial.
Reviewer 3 Report
In the manuscript materials-1061625, the authors developed rGO aerogel to fill in melamine sponge as an oxygen reduction electrocatalyst. There are some interesting findings in the manuscript, and the manuscript can be considered for publication after addressing the following major issues:
- The fitting of deconvoluted XPS N 1s spectrum was a bit arbitrary. No evidence to support the underlying four small peaks. Please redo the fitting.
- The potential reported herein should be calibrated to reversible hydrogen electrode for easy comparisons with the literatures.
- The bare melamine sponge without aerogel should be tested for the ORR under the same test conditions (Figure 5). Please add Tafel plots for GC, MS (GC), and rGOA-MS (GC).
- The authors are encouraged to provide SEM image of melamine sponge for comparison in Figure 1.
- J. Mater. Chem. A should be the journal name instead of J. Mater. Chem. A Mater. Energy Sustain. in the references. Please double check this issue.
Round 2
Reviewer 1 Report
Thank you. I recommend the publication of the article in the present form.
Reviewer 3 Report
The revision made by the authors is satisfactory. The manuscript can be considered for publication.